# Medical and health students' insights into mycetoma: A survey-based study on knowledge and clinical practices

Ali Awadallah Saeed[1,2]*, Samira M. E. Hussein[2], Ahmad Mohammad Al Zamel[3], Lamya Bashir Eisa[4], Azizah Omer Abubaker Daud[5], Hozifa Seedahmed Mukhtar Seedahmed[5], Fatima Omer Ibrahim Ahmed[5], Tho Alyazan Khalil Taher Al-Jabali[5], Eman Kheir[6], Ahmed Hassan Fahal[1]

1 The Mycetoma Research Center, University of Khartoum, Khartoum, Sudan, 2 Faculty of Pharmacy, National University-Sudan, Khartoum, Sudan, 3 Faculty of Medicine and Surgery, Al-Neelain University, Khartoum, Sudan, 4 Faculty of Clinical and Industrial Pharmacy, Hospital Pharmacy, National University, Khartoum, Sudan, 5 The National University, Khartoum,Sudan, 6 Department of periodontology, Faculty of Dental Medicine and Surgery, Associate Professor of Periodontology and Oral Medicine, Centre for Professional Development, National University, Khartoum, Sudan

* aliawadsaeed@nu.edu.sd, alimhsd@gmail.com

## Abstract

### Background

Mycetoma poses significant public health challenges, especially in resource-limited settings. Well-trained healthcare professionals with both comprehensive knowledge and practical skills are essential in combating this disease. Recognising this need, the present study aimed to assess the knowledge and practices of medical and health students in Sudan regarding mycetoma.

### Methods

A cross-sectional survey was conducted among 547 students from various medical and health-related faculties across Sudan. A structured questionnaire assessed participants' awareness, knowledge (epidemiology, causative agents, clinical features, diagnosis, and management), and practical experiences related to mycetoma. The levels of knowledge and practice were categorised as good, moderate, or low. Associations between academic level, faculty type, and performance were analysed.

### Results

A majority of students (80.6%) had heard of mycetoma, indicating a generally high level of awareness. However, only 26.3% demonstrated good knowledge, while 34.6% had moderate and 39.1% had low knowledge. Practical competencies were also limited, with 17.4% showing good practice, 22.7% moderate, and 60% low practice. Senior students and those enrolled in medical faculties exhibited better

**Data availability statement:** All data available within manuscript.

**Funding:** The author(s) received no specific funding for this work.

knowledge and practices compared to junior students. Notably, discrepancies between theoretical knowledge and actual practice were evident, suggesting that current training methods may not sufficiently prepare students for real-world management of mycetoma. Limited clinical exposure, insufficient community-based training, and inadequate emphasis on practical skills were identified as key contributing factors.

## Conclusion

This study highlights critical gaps in both knowledge and practical skills related to mycetoma among Sudanese medical and health students. Equipping future healthcare providers with better skills will strengthen mycetoma diagnosis, management, and control, improving outcomes and reducing its burden in endemic regions. Additionally, the validated assessment tool developed in this study offers a resource for evaluating knowledge and practices related to other neglected tropical diseases.

### Author summary

Mycetoma is a neglected disease that mainly affects people in Sudan, especially in areas with limited healthcare. To better fight this disease, future healthcare workers need to be well-trained, both in understanding the disease and in how to treat it effectively. This study examined over 500 medical and health students across Sudan to assess their knowledge of mycetoma and their preparedness to manage it. While most students had heard of mycetoma, many only knew basic facts and lacked detailed understanding of how the disease spreads, its symptoms, and how to diagnose and treat it. Additionally, although some students practiced good clinical skills, many felt unconfident and had limited hands-on experience. More advanced students and those in medical schools generally knew more and performed better. The study highlights that current training methods need improvement, particularly by providing more practical experience and community-based learning, so that future healthcare workers can better identify and manage mycetoma early. Strengthening education in this way can help reduce the disease's impact, improve patient care, and support efforts to control mycetoma in Sudan.

## Introduction

Mycetoma is a chronic, progressive, and destructive granulomatous infection that primarily affects subcutaneous tissues, with a propensity to invade deeper anatomical structures, including the fascia, muscles, and bones [1,2]. This disfiguring condition is characterised by the formation of painless masses, multiple sinus tracts, and the discharge of macroscopic aggregates of pathogens known as grains [3–6]. The disease

is aetiologically classified into two distinct forms: actinomycetoma, caused by filamentous aerobic bacteria (actinomycetes), and eumycetoma, caused by fungi (eumycetes) [7,8]. Over 70 microbial species have been implicated in its pathogenesis, including *Nocardia brasiliensis, Streptomyces somaliensis, Madurella mycetomatis,* and others, each contributing to variations in clinical presentation, treatment response, and prognosis [1,9–11]. Actinomycetoma, for instance, often progresses more rapidly and responds better to antibiotics, whereas eumycetoma tends to be indolent, drug-resistant, and frequently necessitates surgical intervention [12–14].

Mycetoma exhibits a striking geographical predilection for tropical and subtropical regions, particularly within the "mycetoma belt", a band encompassing arid and semi-arid zones across Africa, Asia, and Latin America [10,15–24]. Environmental factors, such as soil composition, seasonal rainfall, and traditional agricultural practices in rural communities, contribute to its endemicity [7,9,11,25]. Sudan, Mexico, Senegal, Somalia, Mali, and India report the highest prevalence, with Sudan accounting for approximately 70% of global cases [26–28]. This disproportionate burden reflects the confluence of socioeconomic vulnerabilities, climatic conditions, and limited healthcare infrastructure in these regions [29–31].

Mycetoma is typically acquired when the causative organism enters the body through minor skin injuries like thorn pricks or other forms of penetrating trauma. Individuals who walk barefooted and those involved in manual labor, such as agricultural work, are at higher risk due to increased exposure to soil and potential for injury [8].

The insidious progression of mycetoma, coupled with delayed diagnosis due to limited awareness and diagnostic resources, often results in catastrophic outcomes [32–38]. Early-stage lesions are frequently misdiagnosed as benign tumours, foreign body granuloma, or other chronic skin conditions, allowing the infection to advance unchecked [3,39–41]. Over the years, this has led to severe deformities, irreversible disabilities, stigma and systemic complications such as secondary bacterial infections or systemic sepsis [42–44]. In endemic regions, mycetoma perpetuates a vicious cycle of poverty and marginalisation. Rural, low-income populations, particularly subsistence farmers, herders, and labourers, are disproportionately affected due to occupational exposure and limited access to healthcare [29–31,45]. Late-stage disease renders individuals unable to work, driving loss of livelihood, social stigmatisation, and psychological trauma [42,46]. Young adults face exclusion from employment and marriage prospects, while children often drop out of school due to disability, stigma, or familial poverty [2,33].

In 2016, the World Health Organization (WHO) designated mycetoma as a Neglected Tropical Disease (NTD), a landmark decision that bolstered global advocacy, research funding, and policy prioritisation [47–49]. Despite this progress, critical gaps persist in understanding its epidemiology. Reliable data on incidence, prevalence, and spatial distribution remain scarce, hindered by underreporting, fragmented health systems, and the absence of standardised surveillance mechanisms in endemic countries [48,49]. Furthermore, diagnostic challenges, such as the reliance on invasive biopsies for histopathological confirmation and limited access to molecular or imaging technologies, compound delays in treatment [50,51].

Sudan, the global epicenter of mycetoma, epitomises the disease's intersection with socio-medical inequities [52,53]. Over 12,000 registered cases have been reported at the country's Mycetoma Research Centre in Khartoum, although the actual numbers are likely far higher [54]. Most patients originate from remote, resource-poor villages in Sudan's Sennar, White Nile and Gezira states, where traditional barefoot farming in contaminated soil is commonplace [54,55]. Poverty, illiteracy, and cultural beliefs attributing the disease to supernatural causes drive reliance on traditional healers, delaying medical care by years [56]. Even when patients reach specialised centres, financial constraints, drug shortages, and the complexity of long-term therapies often requiring months of antibiotics or costly and ineffective antifungals hinder recovery [47–49,57]. While Sudan has pioneered research and established dedicated treatment centers, systemic barriers, including underfunded health systems, urban-rural healthcare disparities, and political instability, stifle broader progress.

Medical and health students, as future frontline healthcare providers in endemic regions, play a critical role in breaking this cycle. However, gaps in their knowledge and clinical preparedness may undermine early diagnosis, effective management, and patient education [54,58]. This study assesses the current understanding and practices related to mycetoma

among medical students in Sudan, aiming to identify critical deficiencies in recognising early symptoms, differentiating between bacterial and fungal aetiologies, and applying diagnostic or therapeutic protocols. By elucidating these gaps, the findings will inform targeted interventions, such as enhanced curricula, community-based training, and digital learning tools, to equip emerging healthcare workers with the necessary skills to combat this neglected yet devastating disease. Addressing these educational shortcomings is not only a medical imperative but a moral one, integral to alleviating the socioeconomic ravages of mycetoma in vulnerable populations.

## Materials and methods

### Ethics statement

Ethical clearance for the study was obtained from the Faculty of Pharmacy, National University Institutional Review Board, numbered NU-REC/11–018/12, and all participants gave written informed consent in accordance with the Declaration of Helsinki.

### Study design

This descriptive cross-sectional survey was conducted from April 15 to June 15, 2024, targeting undergraduate medical and health students from both public and private universities across Sudan. Students who declined participation or lacked internet access were excluded. A multistage sampling technique was used. First, universities were stratified into public and private categories. Within each category, the intended approach was random selection; however, due to the ongoing conflict in Sudan, the final choice of institutions was constrained by accessibility and contact availability. This resulted in a mixed sampling approach, combining stratification with elements of convenience sampling to ensure broad institutional representation under challenging circumstances. Students from the selected universities were then systematically sampled. This approach aimed to ensure a diverse and representative sample, thereby enhancing the study's reliability and generalisability. To facilitate the distribution of questionnaires, focal persons from each university category were contacted and assisted in disseminating the survey among their peers through stratified sampling.

### Study particpants

The calculated sample size was 384 participants; however, 547 participants were ultimately included in the study.

The simplest formula Cochran formula [59] for unknown population was used which allows for calculating an ideal sample size given a desired level of precision, desired confidence level, and the estimated proportion of the attribute present in the population.

N= PQZ2/d2 was used, where N=sample size, P= prevalence factor, Q=1-p, Z= constant 95% occurred 1.96, d= desired margin.

$$((1.96)2\ (0.5)\ (0.5))/\ (0.05)2\ =\ 384$$
$$N = 384$$

### Description of questionnare

The questionnaire was developed based on a review of existing literature [1,2,7], drafted in English, and validated by the authors to ensure its suitability and relevance within the Sudanese context. A pilot study was conducted to evaluate face and content validity further. The questionnaire comprised two main sections: the first focused on mycetoma knowledge, divided into four parts, epidemiology (7 questions), the causative agents (7 questions), entry routes (7 questions), and infection sites with clinical features (4 questions), as well as diagnosis, management, and complications (3 questions). The second section addressed practices related to mycetoma (10 questions) and the decision-making practices (8 questions).

## Piloting and validation

Expert researchers reviewed the questionnaire for relevance, accuracy, and validity. Reliability testing using Cronbach's alpha yielded good internal consistency: 0.874 for the knowledge subscale and 0.741 for the practice subscale. A pilot study was conducted to evaluate face and content validity further.

Scoring for knowledge and practice was assigned as 2 points for each correct answer and 0 for incorrect responses. Participants' levels of knowledge and practice were categorised according to Bloom's cutoff criteria:

Knowledge: Low (<16.8), Moderate (16.8–22.3), High (22.4–28)

Practice: Poor (<12), Acceptable (12–15.9), Good (16–20)

Normality of scores was assessed using the Shapiro-Wilk test, scatter plots, and measures of skewness and kurtosis, confirming the parametric assumptions. Correlation analyses examined relationships between knowledge and practice (significance at $p < 0.01$). Variations across faculties and academic year groups were tested using ANOVA (significance at $p < 0.05$). Significant ANOVA results were further explored using post-hoc Tukey tests, with the Bonferroni correction applied to adjust for multiple comparisons.

## Data management and analysis

Data were cleaned, entered into Microsoft Excel, and analysed using SPSS version 27. Descriptive statistics (frequencies, proportions, means, and standard deviations) summarised the data. The validity of the instrument was confirmed through Principal Component Analysis (PCA), which indicated suitability based on high coefficients (>0.4), a Kaiser-Meyer-Olkin (KMO) measure above 0.7, and significant Bartlett's tests.

# Results

The study included 547 students from various faculties and universities, with the following distribution: 68.9% were medical students, 23.4% were pharmacy students, and 7.7% were from dentistry, nursing, or other health-related fields. Females comprised 55.8% of the participants, while males accounted for 44.2%. The average age was 23 years. Most students (42.4%) were in their final year of study.

Regarding awareness, 80.6% had heard of mycetoma, and 50.1% were aware of the Mycetoma Research Centre. Half of the participants reported lectures, internet, social media and research papers as their primary sources of information on mycetoma. Approximately one-quarter had seen mycetoma in relatives, roughly one-third had encountered mycetoma patients in the community, and 50% had not previously seen a case of mycetoma. (Table 1).

## Construct validity analysis

The 24 items comprising the knowledge and practice subscales were subjected to principal components analysis (PCA) to evaluate their underlying structure. Sampling adequacy was confirmed by high Kaiser-Meyer-Olkin (KMO) values of 0.934 for the knowledge subscale and 0.765 for the practice subscale, both exceeding the recommended threshold of 0.6. Bartlett's test of sphericity was statistically significant for both subscales ($p < 0.001$), indicating that the correlation matrices were suitable for factor analysis. The PCA results validated the construct structure of the questionnaire, supporting its use for assessing knowledge and practice levels among participants. Analysis of responses revealed notable knowledge gaps across medical, pharmacy, and other health-related faculties, with certain key domains—such as epidemiology, causative agents, and clinical management—showing particularly low scores (Table 2). These findings underscore the need for targeted educational interventions to address discipline-specific deficiencies.

Analysis of the study population's responses to the mycetoma practice assessment items revealed substantial gaps in practical competencies across medicine, pharmacy, and other health-related faculties (Table 3). Deficiencies were particularly evident in areas such as early clinical recognition of mycetoma, proper sample collection and handling for laboratory

**Table 1. The study population characteristics and their knowledge and practice toward mycetoma (n = 547).**

| General characteristics | | N = 547 [1] |
|---|---|---|
| Age | | 23 ± 2.5 |
| Gender | Female | 305 (55.8%) |
| | Male | 242 (44.2%) |
| Faculty | Medicine | 377 (68.9%) |
| | Pharmacy | 128 (23.4%) |
| | Dentistry | 12 (2.2%) |
| | Nursing | 13 (2.4%) |
| | Other faculties | 17 (3.1%) |
| Academic year | First year | 18 (3.3%) |
| | Second year | 70 (12.8%) |
| | Third year | 62 (11.3%) |
| | Fourth year | 165 (30.2%) |
| | Fifth year | 232 (42.4%) |
| Have you heard about mycetoma | No | 106 (19.4%) |
| | Yes | 441 (80.6%) |
| Source of information | Lectures | 319 (40.6%) |
| | Internet | 195 (24.8%) |
| | Social media | 90 (11.5%) |
| | Research paper | 61 (7.8%) |
| | No sources of information | 75 (9.6%) |
| | Others | 45 (5.7%) |
| Have you heard about the Mycetoma Research Center | No | 264 (48.3%) |
| | I don't know | 0 (0.0%) |
| | Yes | 283 (51.7%) |
| Did you see a mycetoma patient in the relatives | No | 336 (61.4%) |
| | I don't know | 77 (14.1%) |
| | Yes | 134 (24.5%) |
| Did you see a mycetoma patient in the community | No | 281 (51.4%) |
| | I don't know | 73 (13.3%) |
| | Yes | 193 (35.3%) |
| Didn't see a patient with mycetoma | No | 180 (32.9%) |
| | I don't know | 70 (12.8%) |
| | Yes | 297 (54.3%) |

[1] n (%); Mean ± SD.

confirmation, appropriate referral pathways, and application of standard treatment protocols. These gaps suggest that while students may possess theoretical awareness of the disease, many lack the hands-on skills and procedural familiarity necessary for effective real-world management. The variation in practice scores between faculties highlights the influence of curriculum content, clinical exposure, and experiential learning opportunities on students' preparedness to address mycetoma in professional settings.

The obtained data showed that medical students had the highest mean scores in both knowledge and practice subscales, while pharmacy students had the lowest mean scores in both subscales (Table 4).

**Table 2. The study population mycetoma in-depth knowledge assessment (n = 547).**

| Knowledge assessment items | | N = 547 [1] | | | | |
|---|---|---|---|---|---|---|
| | | Responses | Medicine | Pharmacy | Other Related Health Faculties | Total (N/%) |
| K 1 | Mycetoma is a unique neglected tropical disease, endemic in many tropical and subtropical regions like Sudan | Correct responses | 307 | 72 | 25 | 404 (73.9%) |
| | | Incorrect responses | 70 | 56 | 17 | 143 (26.1%) |
| K 2 | Mycetoma causative organisms found in soil enter through the skin wound and sharp thorns | Correct responses | 74 | 47 | 18 | 139 (25.4%) |
| | | Incorrect responses | 303 | 81 | 24 | 408 (74.6%) |
| K 3 | The reason for the late admission of patients to the hospital is the fear of amputation | Correct responses | 330 | 114 | 40 | 484 (88.5%) |
| | | Incorrect responses | 47 | 14 | 2 | 63 (11.5%) |
| K 4 | The reason for the late admission of patients to the hospital is a lack of pain at first | Correct responses | 123 | 65 | 19 | 207 (37.8%) |
| | | Incorrect responses | 254 | 63 | 23 | 340 (62.2%) |
| K 5 | The reason for the late admission of patients to the hospital is that they are visiting native healers | Correct responses | 126 | 57 | 18 | 201 (36.7%) |
| | | Incorrect responses | 251 | 71 | 24 | 346 (63.3%) |
| K 6 | The reason for the late admission of patients to the hospital is the cost of medication | Correct responses | 330 | 111 | 38 | 479 (87.6%) |
| | | Incorrect responses | 47 | 17 | 4 | 68 (12.4%) |
| K 7 | Mycetoma infection complications are deformity, disability and may lead to amputation | Correct responses | 90 | 50 | 20 | 160 (29.3%) |
| | | Incorrect responses | 287 | 78 | 22 | 387 (70.7%) |
| K 8 | The most common site of mycetoma is the foot | Correct responses | 56 | 40 | 9 | 105 (19.2%) |
| | | Incorrect responses | 321 | 88 | 33 | 442 (80.8%) |
| K 9 | The infection appears as: Discharges containing grains | Correct responses | 105 | 54 | 18 | 177 (32.4%) |
| | | Incorrect responses | 272 | 74 | 24 | 370 (67.6%) |
| K 10 | The infection appears as swelling with sinuses | Correct responses | 106 | 47 | 13 | 166 (30.3%) |
| | | Incorrect responses | 271 | 81 | 29 | 381 (69.7%) |
| K 11 | The infection appears as smooth, wet, shiny and dyspigmented skin | Correct responses | 167 | 81 | 18 | 266 (48.6%) |
| | | Incorrect responses | 210 | 47 | 24 | 281 (51.4%) |
| K 12 | Culture and X-ray are fully used in the diagnosis of mycetoma | Correct responses | 135 | 73 | 17 | 225 (41.1%) |
| | | Incorrect responses | 242 | 55 | 25 | 322 (58.9%) |
| K 13 | A combination of surgery and medication is the best choice for mycetoma treatment | Correct responses | 80 | 46 | 13 | 139 (25.4%) |
| | | Incorrect responses | 297 | 82 | 29 | 408 (74.6%) |
| K 14 | Prevention of mycetoma can be done by wearing shoes and gloves while working | Correct responses | 68 | 40 | 12 | 120 (21.9%) |
| | | Incorrect responses | 309 | 88 | 30 | 427 (78.1%) |

[1] n (%).

The study revealed that more than half of the students (60.0%) have poor practice regarding mycetoma, despite 60.9% of them possessing moderate to high knowledge. (Table 5)

**Pearson product-moment correlation coefficient**

Results demonstrated strong positive correlation between the knowledge total scores and practice total scores, $r = 0.737$, $n = 547$, $p < 0.001$, with low level of knowledge associated with low level of practice and (54.3%) of the variance in student's practice score was explained by their knowledge score, with a 95% confidence interval ranging from 0.696 to 0.773.

**ANOVA analysis**

Regarding the knowledge subscale, ANOVA test results indicated statistically significant mean differences among faculties groups and academic year groups. The eta squared statistic for faculties and academic year was (0.051) and (0.202), respectively, indicating moderate and large effect sizes.

**Table 3. The study population's mycetoma practice assessment (n = 547).**

| Practice assessment items | | Responses | Faculties | | | Total |
|---|---|---|---|---|---|---|
| | | | Medicine | Pharmacy | Other Health Facilities | (N/%) |
| P 1 | Counselling a mycetoma patient to avoid specific foods | Correct responses | 154 | 33 | 25 | 212 (38.8%) |
| | | Incorrect responses | 223 | 95 | 17 | 335 (61.2%) |
| P 2 | Counselling a mycetoma patient to take a specific type of food | Correct responses | 137 | 27 | 22 | 186 (34.0%) |
| | | Incorrect responses | 240 | 101 | 20 | 361 (66.0%) |
| P 3 | Counselling a mycetoma patient to no need for a specific type of food | Correct responses | 155 | 40 | 22 | 217 (39.7%) |
| | | Incorrect responses | 222 | 88 | 20 | 330 (60.3%) |
| P 4 | Counselling a mycetoma patient to be isolated from the other people | Correct responses | 113 | 30 | 9 | 152 (27.8%) |
| | | Incorrect responses | 264 | 98 | 33 | 395 (72.2%) |
| P 5 | No advice | Correct responses | 179 | 29 | 17 | 225 (41.1%) |
| | | Incorrect responses | 198 | 99 | 25 | 322 (58.9%) |
| P 6 | Give medication to a mycetoma patient depending on clinical manifestations | Correct responses | 307 | 85 | 29 | 421 (77.0%) |
| | | Incorrect responses | 70 | 43 | 13 | 126 (23.0%) |
| P 7 | Advise a mycetoma patient to go to a diagnostic center | Correct responses | 311 | 101 | 29 | 441 (80.6%) |
| | | Incorrect responses | 66 | 27 | 13 | 106 (19.4%) |
| P 8 | If the doctor advises you to give the mycetoma patient a suitable treatment, you will ask the patient about the type of infection | Correct responses | 54 | 21 | 5 | 80 (14.6%) |
| | | Incorrect responses | 323 | 107 | 37 | 467 (85.4%) |
| P 9 | Suitable treatment is Trimethoprim/sulfamethoxazole for the bacterial infection of mycetoma | Correct responses | 227 | 45 | 18 | 290 (53.0%) |
| | | Incorrect responses | 150 | 83 | 24 | 257 (47.0%) |
| P 10 | Suitable treatment is itraconazole for fungal infection of mycetoma | Correct responses | 248 | 76 | 23 | 347 (63.4%) |
| | | Incorrect responses | 129 | 52 | 19 | 200 (36.6%) |

[1] n (%).

Regarding faculty groups, Post-hoc comparisons using the Tukey HSD test indicated that the mean score for the medical faculties was statistically significantly different from that of the Pharmacy faculties at $p < 0.001$ and from other related health faculties at $p = 0.035$.

Regarding different academic years, Post-hoc comparisons using the Tukey HSD test indicated that the mean score for the first year was statistically significantly different from that of the third year at $p = 0.026$ and from the fourth and fifth years at $p < 0.001$. Another statistically significant difference between the mean score of the second year and (third, fourth and fifth year) at $p < .001$. And last, a statistically significant mean difference was detected between the third and fourth years at $p < .001$.

Regarding the practice subscale, ANOVA test results indicated statistically significant mean differences among faculties groups and academic year groups. The eta squared statistic for faculties and academic year was (0.040) and (0.076), respectively, indicating small and moderate effect sizes.

Regarding faculties groups, Post-hoc comparisons using the Tukey HSD test indicated that the mean score for medical faculties was statistically significantly different from that of pharmacy faculties at $p < 0.001$

Regarding different academic years, Post-hoc comparisons using the Tukey HSD test indicated that the mean score for the first year was statistically significantly different from that of the fourth year ($p = 0.015$) and from the fifth year ($p = 0.011$).

A statistically significant difference was detected between the mean scores of the second and third years, with a $p = 0.021$. The last statistically significant mean difference was detected between the second year and the fourth and fifth years. ($p < .001$).

**Table 4. Knowledge and practice mean scores of medical students toward mycetoma (n = 547).**

| KAP subscales | Medicine | | | Pharmacy | | | Other related health faculties | | | Total | | |
|---|---|---|---|---|---|---|---|---|---|---|---|---|
| | N | Mean | SD | N | Mean | SD | N | Mean | SD | N | Mean | SD |
| Knowledge scores | 377 | 18.13 | 7.27 | 128 | 14.23 | 7.76 | 42 | 15.14 | 7.41 | 547 | 16.99 | 7.58 |
| Practice scores | 377 | 10.00 | 4.96 | 128 | 7.61 | 4.55 | 42 | 9.48 | 5.61 | 547 | 9.40 | 5.01 |

**Table 5. The study population knowledge and practice level toward mycetoma (n = 547).**

| scale | Categories N = 547[1] | | | | | |
|---|---|---|---|---|---|---|
| | Low | | Moderate | | High | |
| Knowledge | 214 | 39.1% | 189 | 34.6% | 144 | 26.3% |
| Practice | 328 | 60.0% | 124 | 22.7% | 95 | 17.4% |

[1]n (%).

## Discussion

This study provides valuable insights into the current state of knowledge and practices regarding mycetoma among a group of medical and health students in Sudan, a region where the disease remains endemic and continues to pose significant public health challenges [10]. Sudan is recognised as one of the countries most affected by mycetoma, with numerous cases reported annually, often leading to severe morbidity, disability, and social stigma [42]. Given the critical role that future healthcare professionals play in disease prevention, early diagnosis, and management, understanding their level of awareness and practical skills is essential for developing effective strategies to combat this neglected tropical disease.

The findings of this study reveal both encouraging aspects of awareness and notable gaps that need urgent attention. On the positive side, a substantial proportion of students demonstrated familiarity with mycetoma, indicating that educational efforts, whether through formal curricula, social media or community health initiatives, have begun to raise awareness among future healthcare providers. This baseline knowledge provides a promising foundation upon which targeted educational programmes can be built.

However, despite this general awareness, the study also uncovered significant deficiencies in detailed knowledge about the disease's epidemiology, causative agents, clinical presentation, diagnostic procedures, and management strategies. Such gaps are concerning because they could hinder early detection and appropriate treatment, which are critical for preventing the progression to disfigurement and disability associated with advanced mycetoma cases. Moreover, a lack of comprehensive understanding may contribute to delays in diagnosis, mismanagement, or inadequate patient counselling, ultimately impacting patient outcomes and the overall public health response.

Furthermore, the study highlights that many students lack practical experience and confidence in managing mycetoma cases, underscoring the need for more practical training and community engagement opportunities within medical and health sciences education. Practical skills are vital for effective clinical decision-making, patient education, and community health promotion, key components in controlling diseases that often affect marginalised populations with limited access to healthcare.

In this study, the high percentage (80.6%) of students who have heard of mycetoma indicates a promising baseline awareness among future healthcare providers, which is likely a reflection of their exposure to academic curricula, media campaigns, or personal and community experiences related to the disease. This level of general awareness is

encouraging, as it suggests that the disease is recognised as a health concern within the student population, potentially facilitating early recognition and prompt referral in clinical settings. Additionally, this awareness may be bolstered by public health efforts and community outreach programmes that have raised awareness about the existence of mycetoma and its impact on affected populations.

However, while awareness is an important first step, it is insufficient on its own to ensure effective diagnosis and management of mycetoma. The data indicate that, despite a substantial proportion of students having heard of the disease, their detailed knowledge about critical aspects, such as epidemiology, causative agents, clinical features, diagnostic procedures, treatment options, and potential complications, is variable and, in many cases, limited which could have significant implications for their future clinical practice.

This knowledge gap is concerning because a superficial understanding of mycetoma may result in missed or delayed diagnoses, particularly given that early symptoms can be subtle and easily mistaken for other skin or soft tissue conditions. In endemic regions like Sudan, where the disease often affects vulnerable populations with limited healthcare access, healthcare professionals' ability to recognise the disease early and initiate appropriate management is vital to prevent severe deformities, disabilities, and social stigmatisation [54].

Furthermore, the variability in detailed knowledge among students suggests that current educational strategies may not be sufficiently comprehensive or effective in conveying the complexities of mycetoma. It underscores the need for curriculum enhancements that go beyond mere awareness-raising to include in-depth training on disease epidemiology, risk factors, clinical diagnosis, differential diagnosis, treatment protocols, and management of complications. Such education should also emphasise practical skills, including clinical examination techniques, laboratory diagnostics, and case management, to better prepare students for real-world scenarios.

The disparities in knowledge levels also highlight the importance of targeted interventions, such as interactive workshops, case-based learning, and exposure to patients with mycetoma during clinical rotations, to reinforce theoretical knowledge and translate it into competent clinical practices. Without these improvements, there is a risk that future healthcare providers may overlook early signs of mycetoma or fail to provide optimal care, ultimately impeding efforts to control and reduce the disease burden in endemic regions.

The importance of this gap becomes more apparent when considering the critical role that future physicians and healthcare providers play in controlling and managing neglected diseases, such as mycetoma. Several previous studies have emphasised that inadequate knowledge among healthcare workers contributes to delayed diagnosis, mismanagement, and poor patient outcomes [60]. Therefore, the observed deficiencies in knowledge underscore the urgent need to strengthen medical education regarding mycetoma, especially given its endemicity in Sudan and neighbouring regions [61]. Incorporating dedicated modules, case-based learning, and clinical exposure into the undergraduate curriculum could significantly enhance students' understanding and readiness to handle real-world cases.

In light of the data obtained in this study, there is an urgent need to reevaluate and strengthen the medical and health sciences curricula regarding neglected tropical diseases, such as mycetoma. Incorporating more comprehensive and hands-on training modules, community-based learning, and interdisciplinary approaches can help bridge the identified gaps. Additionally, fostering collaborations with specialised centers, such as the Mycetoma Research Centre in Sudan, can provide students with valuable exposure and mentorship.

Practices related to mycetoma among the students in this study were found to be inconsistent and, in some cases, suboptimal, highlighting a significant gap between theoretical knowledge and practical application. While some students reported engaging in appropriate clinical practices, others exhibited a lack of confidence or demonstrated insufficient hands-on experience in managing mycetoma cases. This disparity highlights a crucial issue: possessing theoretical knowledge alone does not necessarily translate into effective clinical skills or informed decision-making in real-world settings.

One potential reason for this disconnect is limited clinical exposure to mycetoma cases during undergraduate training. In endemic regions like Sudan, although mycetoma is prevalent, opportunities for students to observe, examine, and

manage actual cases may be limited due to resource constraints, lack of specialised clinics, or insufficient integration of community health experiences into the curriculum. As a result, students may learn about the disease theoretically but lack the confidence or competence to identify early signs, differentiate it from other conditions, or implement appropriate management strategies in practice.

Another contributing factor could be the insufficient emphasis on practical skills training related to mycetoma during their education. Traditional teaching methods often prioritise didactic lectures and theoretical assessments, which, while important, do not adequately prepare students for the complexities of clinical decision-making and community-based management. Without structured practical sessions, hands-on workshops, or supervised community health activities focused on mycetoma, students may find it challenging to develop the clinical reasoning and procedural skills necessary for effective diagnosis and management.

Moreover, community engagement initiatives are vital in diseases like mycetoma, which predominantly affect marginalised populations living in remote or underserved areas. However, if students are not actively involved in community outreach programmes or fieldwork that expose them to real-world cases and social determinants of health, their ability to translate classroom knowledge into meaningful practice remains limited. Such experiences are crucial for understanding patient behaviours, cultural beliefs, and barriers to healthcare access, all of which influence disease management and outcomes.

Given the severe consequences of delayed diagnosis and inadequate treatment, such as disfigurement, disability, and social stigma, strengthening practical training and community engagement is essential [47,54]. This can be achieved by integrating clinical rotations focused on mycetoma, organising simulation-based workshops, and establishing partnerships with specialised centers. Additionally, incorporating community health projects into the curriculum will provide students with valuable field experience, improve their clinical skills, and foster a sense of social responsibility.

Bridging the gap between knowledge and practice is critical for improving the management of mycetoma. By enhancing practical training, increasing clinical exposure, and promoting community-based learning, medical and health students will be better prepared to detect early cases, make informed decisions, and implement effective management strategies. Such efforts are vital not only for individual patient outcomes but also for broader public health initiatives aimed at controlling and ultimately eliminating the burden of mycetoma in endemic regions.

The analysis across different academic years and faculties revealed that senior students and those in medical faculties tend to have better knowledge and practices compared to their junior counterparts or students from other health-related disciplines. This pattern underscores the importance of continuous education and reinforcement of key topics throughout the medical and health training pathway. It also emphasises the need for the early integration of neglected tropical diseases, such as mycetoma, into medical and health curricula to ensure that students develop a comprehensive understanding from the outset of their training.

Likewise, the sources of information reported by students, primarily lectures and research papers, highlight the role of formal education in shaping their understanding. However, relying solely on these sources may limit practical exposure and experiential learning. Therefore, expanding opportunities for hands-on experiences, community-based learning, and interactions with patients affected by mycetoma can significantly enhance both knowledge and practice.

The utility of e-learning and social media forums significantly enhances students' knowledge and practical understanding of mycetoma by providing accessible, interactive, and up-to-date educational resources. E-learning platforms offer structured modules, multimedia content, and virtual simulations that facilitate deeper comprehension of the disease's aetiology, diagnosis, and treatment, allowing students to learn at their own pace and revisit complex topics as needed. Meanwhile, social media forums foster real-time discussions, peer-to-peer support, and expert interactions, creating a collaborative environment where students can share insights, ask questions, and stay informed about the latest research and clinical practices related to mycetoma. Together, these digital tools bridge geographical barriers, promote active

engagement, and reinforce theoretical knowledge with practical applications, ultimately improving students' competence and confidence in managing this neglected tropical disease.

This study's limitations include potential selection bias due to participant recruitment during conflict-related constraints, reliance on self-reported data which may introduce recall or social desirability bias, and limited generalisability beyond the surveyed universities.

This study has important implications. Firstly, it identifies critical gaps in current medical education that can be targeted for curriculum reform. Secondly, it emphasises the importance of enhancing practical training and community-based experiences to build confidence and competence among students. Thirdly, by equipping future healthcare providers with better knowledge and skills, the overall health system's capacity to respond to mycetoma can be significantly strengthened, reducing morbidity and transmission.

Additionally, the study contributes a validated assessment tool that can be used in future research to evaluate knowledge and practices related to other neglected tropical diseases. Overall, the findings advocate for integrated educational and public health strategies to combat mycetoma more effectively in Sudan, ultimately improving health outcomes and advancing disease control efforts in endemic areas.

In conclusion, while the study demonstrates that a majority of medical and health students in Sudan have heard of mycetoma, there is considerable room for improvement in their detailed knowledge and practical skills. Bridging these gaps is crucial to strengthening early detection, enhancing patient outcomes, and ultimately contributing to the control of mycetoma in endemic regions. Implementing targeted educational interventions, integrating practical training, and promoting community engagement are critical steps forward. Future research should investigate the specific barriers to knowledge acquisition and skill development, as well as assess the effectiveness of various educational strategies in enhancing the competence of medical and health students in managing mycetoma. Ensuring that future healthcare providers are well-equipped to confront this neglected disease is vital for advancing public health efforts and reducing the burden of mycetoma in Sudan and similar settings.

## Author contributions

**Conceptualization:** Ali Awadallah Saeed, Ahmed Mohammad Al Zamel, Fatima Omer Ibrahim Ahmed, Tho Alyazan Khalil Taher Al-Jabali, Eman Kheir, Ahmed Hassan Fahal.

**Data curation:** Samira M. E. Hussein, Azizah Omer Abubaker Daud, Hozifa Seedahmed Mukhtar Seedahmed, Fatima Omer Ibrahim Ahmed, Tho Alyazan Khalil Taher Al-Jabali, Eman Kheir.

**Formal analysis:** Lamya Bashir Eisa.

**Investigation:** Ali Awadallah Saeed, Ahmed Mohammad Al Zamel, Azizah Omer Abubaker Daud, Hozifa Seedahmed Mukhtar Seedahmed, Tho Alyazan Khalil Taher Al-Jabali.

**Methodology:** Ali Awadallah Saeed.

**Software:** Lamya Bashir Eisa.

**Supervision:** Ahmed Hassan Fahal.

**Validation:** Samira M. E. Hussein, Ahmed Mohammad Al Zamel, Lamya Bashir Eisa.

**Visualization:** Ali Awadallah Saeed, Samira M. E. Hussein, Ahmed Mohammad Al Zamel, Lamya Bashir Eisa.

**Writing – original draft:** Ali Awadallah Saeed, Samira M. E. Hussein, Ahmed Mohammad Al Zamel, Lamya Bashir Eisa, Azizah Omer Abubaker Daud, Hozifa Seedahmed Mukhtar Seedahmed, Fatima Omer Ibrahim Ahmed, Tho Alyazan Khalil Taher Al-Jabali, Ahmed Hassan Fahal.

**Writing – review & editing:** Ali Awadallah Saeed, Ahmed Mohammad Al Zamel, Eman Kheir, Ahmed Hassan Fahal.

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
