## [Decision Letter · Decision Letter 0]

12 Aug 2025

Medical and Health Students’ Insights into Mycetoma: A Survey-Based Study on Knowledge and Clinical Practices

Dear Dr. Saeed,

Thank you for submitting your manuscript to PLOS Neglected Tropical Diseases. After careful consideration, we feel that it has merit but does not fully meet PLOS Neglected Tropical Diseases's publication criteria as it currently stands. Therefore, we invite you to submit a revised version of the manuscript that addresses the points raised during the review process.

Please submit your revised manuscript within 60 days Oct 11 2025 11:59PM. If you will need more time than this to complete your revisions, please reply to this message or contact the journal office at plosntds@plos.org. Please include the following items when submitting your revised manuscript:

We look forward to receiving your revised manuscript.

Kind regards,

Joshua Nosanchuk, MD

Section Editor

Shaden Kamhawi

co-Editor-in-Chief

Paul Brindley

co-Editor-in-Chief

**Reviewers' Comments:**

Reviewer's Responses to Questions

**Key Review Criteria Required for Acceptance?**

**Methods:**

-Are the objectives of the study clearly articulated with a clear testable hypothesis stated?

-Is the study design appropriate to address the stated objectives?

-Is the population clearly described and appropriate for the hypothesis being tested?

-Is the sample size sufficient to ensure adequate power to address the hypothesis being tested?

-Were correct statistical analysis used to support conclusions?

-Are there concerns about ethical or regulatory requirements being met?

Reviewer #1: Full comments have been provided in my overall review of the manuscript

Reviewer #2: (No Response)

**Results:**

-Does the analysis presented match the analysis plan?

-Are the results clearly and completely presented?

-Are the figures (Tables, Images) of sufficient quality for clarity?

Reviewer #1: Full comments have been provided in my overall review of the manuscript

Reviewer #2: (No Response)

**Conclusions:**

-Are the conclusions supported by the data presented?

-Are the limitations of analysis clearly described?

-Do the authors discuss how these data can be helpful to advance our understanding of the topic under study?

-Is public health relevance addressed?

Reviewer #1: Full comments have been provided in my overall review of the manuscript

Reviewer #2: (No Response)

**Editorial and Data Presentation Modifications?**

Reviewer #1: (No Response)

Reviewer #2: (No Response)

**Summary and General Comments:**

Reviewer #1: Abstract

The abstract is very long and should be shortened.

In paragraph one of the abstract, these phrases can be removed: 1. a neglected tropical disease endemic in Sudan, 2. Recognising the importance of these competencies, 3. the future frontline health workers.

The abstract does not provide sufficient information about the study's methods. Include a sentence about the tool/questionnaire used for the study.

The abstract should be rewritten to highlight only the most important findings of the study. Start with a background that has one or two sentences stating the aim of the study and/or why it is important. Summarise the methods into two or three sentences. State the study design, number of participants, the period the study was conducted, and the tool used for the study. The next paragraph should summarise the major results/findings of the study. Just start the results without explaining. Sentences like the ones from lines 41 to 43, and 44-46 are not necessary. Have a final paragraph that concludes the abstract. State the overall finding of the study (line 57-59 looks okay for this) and provide recommendations in one or two sentences based on the main findings.

Lines 62-73 should be deleted from the abstract. This can be included in the conclusion of the main manuscript.

Introduction

The introduction is nicely written, but could you add information about how mycetoma spread or is acquired to the first paragraph?

Materials and methods

Line 190: Explain how the sample size was calculated. Provide the formula and the assumptions made in calculating the sample size.

In line 186, you mentioned that universities were randomly selected, and then in line 194 they were selected based on contact availability (suggesting that they were conveniently selected). Can you clarify this?

Divide the first few paragraphs of the methods into these sections: Study design, 2. Study participants (sample size calculations, inclusion and exclusion criteria, sampling technique), 3. Questionnaire (description of the questionnaire)

The paragraph titled piloting and validation, can be added to the Questionnaire section if it is created.

I think lines 220-224 would best be added to the questionnaire section to explain how the answers were scored and finally classified into good or poor knowledge and practice. You need to explain how you selected the cutoffs for defining low, moderate, and high knowledge, as well as poor, acceptable, and good knowledge.

The methods does not indicate whether ethical approval was obtained for the study.

Results

Remove “I don’t know” from the options for Have you heard of mycetoma.

Change Nurse to “Nursing” on table 1

Kindly provide a table description of tables 2 and 3 that highlights some of the responses.

Is there a reason why the full faculties are presented in Table 1, but in the subsequent tables, only medicine and pharmacy are specified, with the remaining fields grouped into one?

I am somewhat concerned about the questions used to assess both mycetoma knowledge and practice. Regarding knowledge assessment, I am wondering how questions K3, K4, K5, and K6 help define whether an individual knows mycetoma. To assess their knowledge, I thought that you would ask them basic questions about the cause, symptoms, transmission, and treatment/prevention of the disease to see whether they are knowledgeable or not, as you did with the remaining questions.

I need a little explanation about how the practice assessment items help to define good practice regarding mycetoma. Are the questions aimed at understanding whether participants can perform those activities or to know whether they have done those activities before?

I suggest merging tables 4 and 5 to tables 2 and 3.

Please, provide the table associated with the “ANOVA analysis”. I think that if you want to find out how the demographics of the participants are associated with knowledge and practice regarding mycetoma, then reclassify the scores dichotomously into good and poor knowledge/practice. This way, you can use logistic regression (instead of ANOVA) to determine the factors associated with knowledge/practice.

Discussion

The authors stated that a substantial portion of the students demonstrated knowledge of mycetoma. But from Table 4, only 26.3% had high knowledge of mycetoma. This makes me wonder how a good overall knowledge was defined. At what percentage is a good knowledge defined?

Line 370 states that 80.6% of the students have heard about mycetoma. Where is that reported in the results? Again, since knowledge was not assessed with a single question but a set of questions, overall knowledge should be defined by a composite value such as the ones found in Table 4.

Please include the limitations of the study to the manuscript.

Reviewer #2: 1. Abstract needs revision by structuring it and focus on key aspects

2. Describe the sampling technique in figure

3. The introduction very vast and lack focus, why mycetoma only selected for purpose of this study?

4. There is ambiguity is it digitally collected or paper based?

5. what about data quality control issues, not described?

6. How you control confounding effects?

7. Regarding target population there is ambiguity, why medical and other mixed? which year? …

8. The limitations not described

9. how you clearly measure level of awareness, low, moderate, indepth knowledge? clear operational definitions needed

10. How reliability ensured? from validity why only face and content validity only focused

PLOS authors have the option to publish the peer review history of their article (what does this mean? ). If published, this will include your full peer review and any attached files.

**Do you want your identity to be public for this peer review?** For information about this choice, including consent withdrawal, please see our Privacy Policy .

Reviewer #1: No

Reviewer #2: No

**Figure resubmission:**

**Reproducibility:**



---

## [Editor Report · Decision Letter 1]

20 Sep 2025

Dear Dr. Saeed,

We are pleased to inform you that your manuscript 'Medical and Health Students’ Insights into Mycetoma: A Survey-Based Study on Knowledge and Clinical Practices' has been provisionally accepted for publication in PLOS Neglected Tropical Diseases.

Best regards,

Joshua Nosanchuk, MD

Section Editor

Shaden Kamhawi

co-Editor-in-Chief

Paul Brindley

co-Editor-in-Chief

---

## [Editor Report · Acceptance letter]

Dear Dr. Saeed,

We are delighted to inform you that your manuscript, "Medical and Health Students’ Insights into Mycetoma: A Survey-Based Study on Knowledge and Clinical Practices," has been formally accepted for publication in PLOS Neglected Tropical Diseases.

Best regards,

Shaden Kamhawi

co-Editor-in-Chief

Paul Brindley

co-Editor-in-Chief
